**Data Availability Statement:** The file containing the raw data is available from figshare.com (http://doi.org/10.6084/m9.figshare.14413001).

# Hepatitis C continuum of care: Experience of integrative hepatitis C treatment within a human immunodeficiency virus clinic in Indonesia

Evy Yunihastuti [1,2]*, Rahmat Hariyanto[1†], Andri Sanityoso Sulaiman[1], Kuntjoro Harimurti[1]

**1** Department Internal Medicine, Faculty of Medicine Universitas Indonesia/Cipto Mangunkusumo Hospital, Jakarta, Indonesia, **2** HIV Integrated Clinic, Cipto Mangunkusumo Hospital, Jakarta, Indonesia

† Deceased.
* evy.yunihastuti@gmail.com

## Abstract

### Introduction

Direct-acting antiviral drugs (DAAs) have changed the paradigm of hepatitis C therapy for both HCV/HIV co-infected and HCV mono-infected patients. We aimed to describe the HCV continuum of care of HIV-infected patients treated in an HIV clinic after a free DAA program in Indonesia and identify factors correlated with sofosbuvir-daclatasvir (SOF-DCV) treatment failure.

### Methods

We did a retrospective cohort study of adult HIV/HCV co-infected patients under routine HIV-care from November 2019 to April 2020 in the HIV integrated clinic of Cipto Mangunkusumo Hospital, Jakarta, Indonesia. We evaluated some factors correlated with sofosbuvir-daclatasvir treatment failure: gender, diabetes mellitus, previous IFN failure, cirrhosis, concomitant ribavirin use, high baseline HCV-RNA, and low CD4 cell count.

### Results and discussion

Overall, 640 anti-HCV positive patients were included in the study. Most of them were male (88.3%) and former intravenous drug users (76.6%) with a mean age of 40.95 (SD 4.60) years old. Numbers and percentages for the stages of the HCV continuum of care were as follows: HCV-RNA tested (411; 64.2%), pre-therapeutic evaluation done (271; 42.3%), HCV treatment initiated (210; 32.8%), HCV treatment completed (207; 32.2%), but only 178 of these patients had follow-up HCV-RNA tests to allow SVR assessment; and finally SVR12 achieved (178; 27.8%). For the 184 who completed SOF-DCV treatment, SVR12 was achieved by 95.7%. In multivariate analysis, diabetes mellitus remained a significant factor correlated with SOF-DCV treatment failure (adjusted RR 17.0, 95%CI: 3.28–88.23, p = 0.001).

**Funding:** The authors received no specific funding for this work.

**Competing interests:** The authors have declared that no competing interests exist.

## Conclusions

This study found that in the HCV continuum of care for HIV/HCV co-infected patients, gaps still exist at all stages. As the most commonly used DAA combination, sofosbuvir daclatasvir treatment proved to be effective and well-tolerated in HIV/HCV co-infected patients. Diabetes mellitus was significant factor correlated with not achieving SVR12 in this population.

## Introduction

Co-infection of the human immunodeficiency virus (HIV) with hepatitis C virus (HCV) is still one of the world's significant health problems. After successful antiretroviral therapy (ART), chronic viral hepatitis still accounts for about 10% of mortality among HIV-infected patients [1]. In the Southeast Asian population, the prevalence of hepatitis C co-infection in HIV patients is 4.1%, while the prevalence of HIV/HCV co-infection in Indonesia is 17.9% [2].

Before the direct-acting antiviral (DAA) therapy era, HCV cure rates were significantly lower in HIV/HCV co-infected patients than in HCV mono-infected patients [3]. Nowadays, sustained virological response (SVR) rates in HIV-infected people match those in HIV-negative people [4]. DAA treatment has been recommended in many guidelines due to its oral administration, fewer and less severe side effects, and higher SVR rate compared to interferon (IFN) treatment [5–7]. However, several barriers to DAA therapy persist in lower-middle-income countries, including Indonesia. These barriers include limited case findings, high diagnostic and drug pricing, difficulties in defining the liver fibrosis phase, high patient lost to follow up, weak data systems, and insufficient finances. There is limited access to and choice of DAAs for HIV-infected patients [7–9].

The World Health Organization (WHO) aims to eliminate HCV as a public health threat by 2030, with the goal of reducing incidence by 80% and HCV-related mortality by 65% from 2015 numbers [10]. To achieve this, WHO recommends a continuum of care (CoC), which is a framework that describes the successive steps in health care required for individuals to achieve optimal health outcomes. CoC is an essential framework to predict, monitor, and evaluate progress in achieving these targets and allows country or population comparison [11].

Several reports have demonstrated attempts to achieve HCV micro-elimination in HIV/HCV co-infected patients within HIV clinics [12–14]. Compared to HCV mono-infected patients, HIV/HCV co-infected patients are more likely to be in care and have better access to HCV treatment [14].

The Indonesian government launched a free DAA program in seven demonstration provinces in 2017, using mainly a combination of sofosbuvir (SOF) and daclatasvir (DCV) [7.8]. Our hospital was one of the first sites of this program. This study was carried out in HIV integrative clinic of a university hospital in Jakarta. The main goal was to describe the HCV continuum of care among HIV-infected patients after the introduction of the free DAA program. An additional goal was to identify factors correlated with not achieving SVR at week 12 (SVR12) of SOF-DCV, the most common drug combination used in Indonesia.

## Methods

### Study population

We conducted a retrospective cohort study of HIV/HCV co-infected patients in the HIV integrated clinic of Cipto Mangunkusumo Hospital. This clinic provides comprehensive HIV care,

including opportunistic infection, mental health, and hepatitis care. Hepatologists have been providing DAA treatment once a week since the free DAA program started. The multidisciplinary team was formed in response to the initiation of the DAA program in 2017 included HIV physicians, counselors, nurses, pharmacy, and data support staff. The available drugs are sofosbuvir, daclatasvir, simeprevir (SMV), ribavirin (RBV, and elbatasvir-grazoprevir (EBR-GZR). Elbatasvir-grazoprevir is provided for dialysis patients. HCV genotyping test is not a requirement of starting DAA treatment. All HCV/HIV co-infected patients are eligible for DAA treatment regardless of their fibrosis status. For those with severe opportunistic infections or those using rifampicin for tuberculosis treatment, treatment is delayed until the condition is resolved or antituberculosis is stopped.

All HIV-infected patients, aged 18 years and older, and known to be anti-HCV positive, who were following routine HIV-care from November 2019 to April 2020, were included in the cascade of care evaluation. Patients who started taking sofosbuvir and daclatasvir in 2017–2019 were included for the analysis of to evaluate factors correlated with treatment failure. This additional analysis excluded pregnant women and hepatocellular carcinoma patients.

## Data collection

Data was primary derived from medical records, included demographics, transmission risk, and antiretroviral therapy. Three core indicators were used to monitor and evaluate the global health sector strategy on viral hepatitis B and C based on WHO recommendations. These included: (1) testing, (2) treatment, and (3) cure. They were modified and studied as HCV continuum of care as follows. The entry port into the cohort was HCV antibody positivity, and then several steps in the continuum of care were assessed: Step 1: HCV-RNA testing; Step 2: complete pre-therapeutic eligibility evaluation, including liver fibrosis assessment; Step 3: treatment initiation; Step 4: treatment completion; Step 5: sustained virological response at week 12 [12].

The potential factors correlated with SOF-DCV treatment failure, evaluated based on various DAAs studies, were gender, diabetes mellitus, previous IFN failure, cirrhosis, concomitant use of ribavirin (RBV), high baseline HCV-RNA (more than 800,000 IU/mL, 800,000 IU/mL or below), and low CD4 cell count (below 200 cells/mm$^3$, a minimum of 200 cells/mm$^3$) [15–20]. With this combination, non-cirrhotic patients got 12 weeks of treatment, and cirrhotic patients got 24 weeks. The daclatasvir dose was 90 mg in those using efavirenz or nevirapine-based ART and 60 mg in those using other regimens. Detectable HCV-RNA 12 weeks after treatment completion was defined as not achieving sustained virologic response (SVR12) or treatment failure. Ethical approval was provided by the Ethics Committee of the Faculty of Medicine Universitas Indonesia for the use of routinely collected anonymous data with a waiver for informed consent.

## Statistical analysis

Results are presented as proportions. The proportion (%) of patients involved in each step of the care cascade was calculated by comparing it to the starting population. Bivariate analysis using the Chi-squared test was used to identify factors associated with not achieving SVR12.

Independent factors correlated with SOF-DCV treatment failure were assessed using a multivariate logistic regression, which included gender, CD4 cell count, diabetes mellitus, history of IFN failure, baseline HCV-RNA, cirrhosis, and ribavirin combination. A backwards stepwise selection process was used to include factors with the significance level of P < 0.25 for the final model. The level of significance was set at 5%, and the risk ratio (RR) with 95% CI was

calculated to determine the association. A p-value of less than 0.05 was considered statistically significant.

## Results and discussion

### Patients

During the study period, a total of 2094 HIV-infected patients were in routine care. Of these patients, 664 were HIV/HCV co-infected (31.8%), all of which used antiretroviral therapy. Twenty-four patients (3.6%) had successfully been treated with an IFN-based regimen before the availability of DAA, leaving 640 patients for further analysis. Most of them were male (565 patients; 88.3%), with a mean age of 40.95 (SD 4.60) years old. HIV and HCV transmission routes were as follows: 490 (76.6%) former intravenous drug users, 98 (15.3%) heterosexual contact, 8 (1.3%) homosexual contact, 3 (0.3%) other routes, and for 41 (6.4%) unknown transmission route.

### HCV treatment cascade

Fig 1 shows the diagnosis-based HCV care cascade since the initiation of the free DAA program in July 2017. In total, 411 of 640 (64.2%) anti-HCV-positive patients underwent HCV-RNA testing, of whom 359 patients were HCV-RNA positive (87.3%). Access to affordable HCV-RNA tests was identified as a major pitfall to start the treatment process as HCV-RNA quantification is still considered an expensive test for most of these patients. The same gaps in the proportion of patients who screened as anti-HCV antibody positive but did not get a confirmatory viral load test were also reported in some countries [7]. The usual cost for the HCV-RNA test in Indonesia is USD 139 [21]. When the government launched the free DAA program, HCV-RNA reagent was included in the coverage. However, supply did not match the number of anti-HCV-positive patients. Therefore, clinicians might have prioritized HCV-RNA tests for patients who were considered ready for DAA treatment. Some of these patients still had severe opportunistic infections or were using rifampicin for tuberculosis treatment which is known to interact with most available DAA drugs.

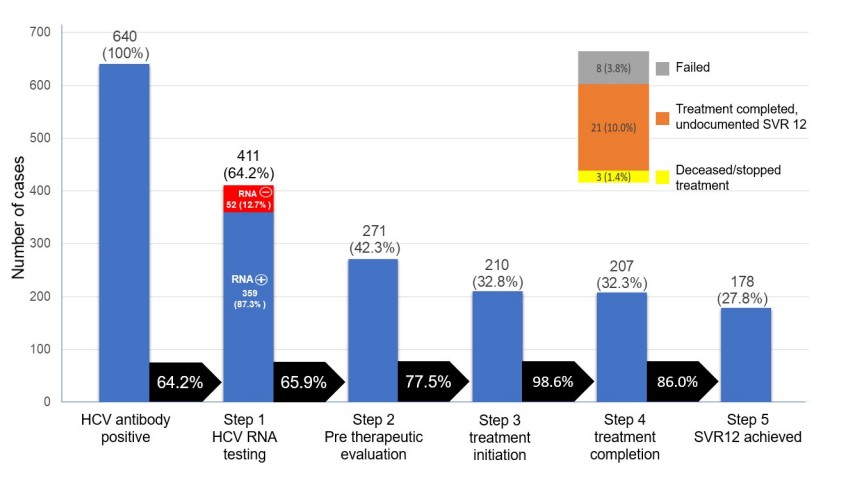

**Fig 1. HCV care cascade for HCV diagnosed HIV-infected patients in care.** Arrows represent the proportion of patients in the previous stage of the cascade that progressed to the next stage of care. Note: SVR = sustained virological response.

Of the 359 HCV-RNA positive patients, 271 patients (75.5%) started their pre-treatment evaluation to define the level of fibrosis, and 210 patients started DAA treatment (58.5%). One possible reason for not doing the pre-treatment evaluation is the cost of the diagnostic procedure before starting DAA. Transient elastography (TE) still carries a high cost, thus often causing a delay in treatment initiation. In the previous report, we proposed an APRI score $\geq$ 1 or a FIB-4 score $\geq$ 1.66 as an alternative to TE for defining cirrhosis in HIV/HCV co-infected patients before starting DAA [9].

Due to the free DAA program's interruption, the remaining 61 patients who had pre-treatment evaluation could not start treatment in 2020. While waiting for the availability of the drugs, four of the patients passed away. The free program started again in January 2021, and it is very likely that some of these patients will initiate DAA treatment soon.

Patients were treated with the following DAA combination: SOF-DCV (98.5%), SOF-DCV-RBV (0.5%), SOF-SMV (0.5%), and EBR-GZR (0.5%). Almost all patients (98.6%) who started DAA completed their treatment, in either a 12 or 24-weeks duration. Three patients did not complete the treatment course: two patients passed away and one stopped treatment due to side effects.

Of the 207 patients who completed the treatment, 178 achieved SVR12 (86.0%), and eight failed to do so. Eight percent of these patients (21 patients) did not have HCV-RNA test 12 weeks after treatment completion. The disruption of the program, including disruption to the supply of free HCV-RNA reagent, was one of the reasons. This point of cascade constitutes the biggest gap in several countries. A study describing the cascade of care of all HCV-infected patients in Cambodia, India, Vietnam, Rwanda, Myanmar, Nigeria, and Indonesia has shown a gap of 55% between initiation of treatment and SVR12 HCV-RNA conducted [7]. However, this study presents better SVR12 documentation as the gap between treatment initiation and SVR12 documentation was only 10%.

To the authors' knowledge, this is the first study to provide an assessment of cascade of care for HCV among HIV-infected patients in Indonesia after the introduction of the free DAA program. Compared to the Indonesian cascade of care for all HCV-infected patients, the cohort described in this study exhibited a different pattern. Both have shown similar treatment initiation rates: 32.8% of 640 HCV antibody positive patients in this study started DAA treatment while 31.8% of 17200 HCV antibody positive patients in the Indonesian report started DAA. The SVR12 rate in our HCV care cascades was 27.8% (178 of 640 HIV/HCV co-infected patients), higher than the SVR12 rate documented in national HCV care cascades (9.5%: 1635 of 17200 all HCV-infected patients) [7]. Our DAA SVR12 treatment rate was considered lower than studies reported in other countries [12, 13, 22–24].

Regarding safety and tolerability, the main adverse effects recorded during and after 12 weeks of treatment cessation were headache (20%), fatigue (18.6%), nausea (7.6%), itchiness (4.8%), gastrointestinal trouble (3.3%), muscle pain (1.9%), sleep disturbance (1.4%), and fever (1.4%). Only one person stopped treatment due to headaches. Others only experienced mild and well-tolerated effects.

This study shown that integrating HCV services within an HIV clinic has overcome several barriers to HCV treatment care, such as referral systems and treatment adherence. The multidisciplinary team that communicated regularly had promoted better engagement in the HCV treatment care, although there was an interruption of drug supply later on. The SVR12 rate in HIV/HCV-coinfected patients in this clinic exceeds the rated reported in national data and shows the effectiveness of this model [7]. Several studies have shown similar efforts to eliminate HCV infection within HIV clinics. Rizk, et al described a co-located HCV clinic within a hospital-based HIV clinic in Connecticut. Of 173 patients, 70.5% initiated DAA treatment resulting in 56.1% SVR12. Most of these patients, however, had mental health issues, abused

alcohol, and were active drug users [12]. Though 76.6% of patients in this study were previously intravenous drug users, we did not have data on currently active drug users. Other studies have also shown successful efforts toward HCV micro elimination in HIV care after universal access to DAAs [12, 13, 25–27].

## Factors correlated with SOF-DCV treatment failure

Overall, 184 HIV/HCV co-infected patients completed SOF-DCV treatment and underwent HCV-RNA quantification. These patients had already been using antiretroviral treatment for a median of 9 (5–12) years, mainly NVP-based therapy (43.5%) and EFV-based therapy (41.8%), as seen in Table 1. Most of the patients had an absolute CD4 cell count of more than 200 cells/mm$^3$.

SVR12 for patients using SOF-DCV regimen was achieved by 176 of 184 patients (95.7%). This SVR rate was comparable with rates reported in phase III studies [28], and other real-life situations in HIV/HCV co-infection [15, 16, 29, 30], even though HCV genotyping was not a

**Table 1. Characteristics of patients completed sofosbuvir-daclatasvir treatment cascade (n = 184).**

| Characteristics | |
|---|---|
| **Male, n (%)** | 165 (89.7%) |
| **Age in years, mean (SD)** | 38.1 (4.2) |
| **HIV/HCV risk factor, n (%)** | |
| History of IVDU | 166 (90.2%) |
| Heterosexual transmission | 15 (8.2%) |
| Homosexual transmission | 3 (1.6%) |
| **Diabetes mellitus, n (%)** | 9 (4.9%) |
| **Cirrhosis, n (%)** | 38 (20.7%) |
| **Previous IFN treatment failure, n (%)** | 15 (8.2%) |
| **HCV treatment regimen, n (%)** | |
| SOF + DCV (90) | 147 (79.9%) |
| SOF + DCV (60) | 36 (19.6%) |
| SOF + DCV (90) + RBV | 1 (0.5%) |
| **SOF-DCV duration, n (%)** | |
| 12 weeks | 143 (77.7%) |
| 24 weeks | 41 (22.3%) |
| **Baseline HCV-RNA, n (%)** | |
| ≥ 800.000 IU/mL | 140 (76.1%) |
| < 800.000 IU/mL | 44 (23.9%) |
| **ART duration in years, median (IQR)** | 9 (5–12) |
| **ART used, n (%)** | |
| NVP-based | 80 (43.5%) |
| EFV-based | 77 (41.8%) |
| LPV/r based | 25 (13.6%) |
| Other | 2 (1.1%) |
| **Recent absolute CD4 cell, n (%)** | |
| <200 cells/mm$^3$ | 26 (14.1%) |
| ≥200 cells/mm$^3$ | 158 (85.9%) |

SD: Standard deviation; IVDU: Intravenous drug user; IFN: Interferon; SOF: Sofosbuvir; DCV: Daclatasvir; RBV: Ribavirin; ART: Antiretroviral therapy; IQR: Interquartile range; NVP: Nevirapine; EFV: Efavirenz; LPV/r: Lopinavir/ritonavir.

requirement of starting this treatment program. A recent meta-analysis has shown that the SVR12 rate of SOF-DCV±RBV among HIV/HCV co-infected patients in clinical-trials was 97% (95% CI 93–99%), and in real-world studies 94.1% (95% CI 91.2–96.4%) [4].

Eight patients (4.3%) failed to achieve SVR12 and needed further treatment. Treatment failure was not defined by their gender, recent CD4, previous IFN treatment failure, RBV combination, nor higher baseline HCV-RNA before starting SOF-DCV treatment, as seen in Table 2. In multivariate analysis, the presence of cirrhosis could not predict treatment failure, although it was significant in univariate analysis. In this study, nine patients had been diagnosed with diabetes mellitus before starting SOF-DCV treatment. One-third of these 9 patients did not achieve SVR12, showing that. patients with diabetes mellitus had a 17 times higher treatment failure rate (95% CI 3.28–88.23). A study by Patel, et al supported this finding. In that study, diabetes mellitus was negatively associated with achieving SVR12 (OR 0.68, 95% CI 0.13–0.94) [15]. A prior study from Spain of 1059 HCV mono-infected patients in the interferon era showed that insulin resistance, the driving factor that leads to type 2 diabetes, was a negative predictor for achieving SVR12 (OR 0.44, 95% CI 0.20–0.97) [31].

For those who did not achieve SVR12, testing for HCV genotype might be needed for future treatment options. In the original phase III ALLY-3 study, a much lower SVR12 rate was observed in genotype 3-infected patients with cirrhosis treated with SOF-DCV for 12 weeks [28]. The addition of RBV to SOF-DCV was recommended to improve the SVR12 response in genotype 3 infected patients with cirrhosis [32, 33]. Our previous study in 2014 showed that genotype 3 is the second most common HCV (21.9%) genotype in HIV/HCV co-infected patients after genotype 1 (68.6%) [34].

**Table 2. Results from univariable and multivariable logistic regression models to identify independent factors correlated with sofosbuvir and daclatasvir treatment failure (n = 184).**

| Factors | Non-SVR12 | SVR12 | RR (95%CI) | p | Adjusted RR (95% CI) | p |
|---|---|---|---|---|---|---|
| | N (%) | N (%) | | | | |
| **Gender** | | | | | | |
| Male | 8 (4.8%) | 157 (95.2%) | - | 1.000 | | |
| Female | 0 (0%) | 19 (100%) | | | | |
| **CD4** | | | | | | |
| <200 cells/mm$^3$ | 2 (7.7%) | 24 (92.3%) | 2.03 (0.43–9.50) | 0.315 | | |
| ≥ 200 cells/mm$^3$ | 6 (3.8%) | 152 (96.2%) | | | | |
| **Diabetes mellitus** | | | | | | |
| Yes | 3 (33.3%) | 6 (66.7%) | 11.67 (3.29–41.33) | 0.004 | 17 (3.28–88.23) | 0.001 |
| No | 5 (2.9%) | 170 (97.1%) | | | | |
| **History of IFN failure** | | | | | | |
| Yes | 0 (0%) | 15 (100%) | - | 1.000 | | |
| No | 8 (4.7%) | 161 (95.3%) | | | | |
| **Baseline HCV-RNA** | | | | | | |
| ≥ 800.000 IU/mL | 7 (5.0%) | 133 (95.0%) | 2.20 (0.28–17.40) | 0.682 | | |
| < 800.000 IU/mL | 1 (2.3%) | 43 (97.7%) | | | | |
| **Cirrhosis** | | | | | | |
| Yes | 4 (10.5%) | 34 (89.5%) | 3.84 (1.01–14.66) | 0.058 | 1.95 (0.34–11.04) | 0.451 |
| No | 4 (2.7%) | 142 (97.3%) | | | | |
| **Ribavirin combination** | | | | | | |
| No | 8 (4.4%) | 175 (95.6%) | - | 1.000 | | |
| Yes | 0 (0%) | 1 (100%) | | | | |

SVR: Sustained virologic response; IFN = interferon

This study has several limitations. First, not all treated patients have HCV-genotyping data. Though genotype 3 was the second most prevalent in this population, the percentage of cure for those who completed treatment was more than 90%. Second, the study included a relatively small number of patients with HCV-RNA evaluation after treatment completion, thus limiting the number of variables analysed by multivariate analysis to find factors correlated with treatment failure. This number has been an obstacle to making a firm conclusion. Furthermore, HCV reinfections after DAA treatment and cures after retreatment were not evaluated in this study. Finally, as this is a single tertiary center experience with a selected population of young age, mostly previous IVDU, low cirrhotic proportion, and mostly interferon treatment-naive, these results may not be generalized to other health care systems in different regions.

## Conclusions

Establishing HCV treatment within a HIV clinic facilitated a better HCV care cascade of HIV/HCV co-infected patients. We found sofosbuvir-daclatasvir treatment to be an effective treatment in HIV/HCV co-infected patients, which resembled other real-world studies, and supports the use of DAA to achieve HCV elimination in 2030. Diabetes mellitus was significant factor correlated with not achieving SVR12 in this population.

## Acknowledgments

We are grateful for the support from Fhadilla Amelia, Veritea Natali, Indah Mediana, all attending staffs, physicians, pharmacists, and nurses of HIV Integrated Clinic Cipto Mangunkusumo Hospital, Jakarta. We did not receive any grant for this study.

This paper was dedicated to co-author and our student, Rahmat Hariyanto, who fought courageously against malignant cancer. Rahmat Hariyanto passed away before the submission of the final version of this manuscript. Evy Yunihastuti accepts responsibility for the integrity and validity of the data collected and analyzed by the deceased author.

## Author Contributions

**Conceptualization:** Evy Yunihastuti, Rahmat Hariyanto, Andri Sanityoso Sulaiman, Kuntjoro Harimurti.

**Data curation:** Evy Yunihastuti, Rahmat Hariyanto, Andri Sanityoso Sulaiman.

**Formal analysis:** Rahmat Hariyanto.

**Funding acquisition:** Evy Yunihastuti.

**Investigation:** Evy Yunihastuti, Rahmat Hariyanto.

**Methodology:** Evy Yunihastuti, Rahmat Hariyanto, Kuntjoro Harimurti.

**Resources:** Evy Yunihastuti, Andri Sanityoso Sulaiman.

**Software:** Rahmat Hariyanto.

**Supervision:** Evy Yunihastuti.

**Validation:** Evy Yunihastuti, Kuntjoro Harimurti.

**Visualization:** Evy Yunihastuti, Rahmat Hariyanto.

**Writing – original draft:** Evy Yunihastuti, Rahmat Hariyanto.

**Writing – review & editing:** Evy Yunihastuti, Andri Sanityoso Sulaiman, Kuntjoro Harimurti.

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
