## [Decision Letter · Decision Letter 0]

19 May 2021

PONE-D-21-12487

Hepatitis C Continuum of Care: an Experience of Integrative Hepatitis C Treatment within Human Immunodeficiency Virus Clinic in Indonesia

PLOS ONE

Dear Dr. Yunihastuti,

Thank you for submitting your manuscript to PLOS ONE. After careful consideration, we feel that it has merit but does not fully meet PLOS ONE’s publication criteria as it currently stands. Therefore, we invite you to submit a revised version of the manuscript that addresses the points raised during the review process.

Your manuscript was reviewed by 2 experts in the field. Both identified many important problems in your submission, which require your attention. Please review the attached comments and provide point-by-point responses.

We look forward to receiving your revised manuscript.

Kind regards,

Yury E Khudyakov, PhD

Academic Editor

PLOS ONE

Journal Requirements:

Reviewers' comments:

Reviewer's Responses to Questions

**Comments to the Author**

1. Is the manuscript technically sound, and do the data support the conclusions?

Reviewer #1: Partly

Reviewer #2: Partly

2. Has the statistical analysis been performed appropriately and rigorously? 

Reviewer #1: I Don't Know

Reviewer #2: Yes

3. Have the authors made all data underlying the findings in their manuscript fully available?

Reviewer #1: No

Reviewer #2: Yes

4. Is the manuscript presented in an intelligible fashion and written in standard English?

Reviewer #1: No

Reviewer #2: Yes

5. Review Comments to the Author

Reviewer #1: 1. The paper provides a good account of a Hepatitis C treatment program in Indonesia. The data presented is good. The authors will however need to revise their manuscript with the help of an editor or a native English speaker to improve the grammar and sentence structure to bring the manuscript up to the standards required for publication.

2. In the second paragraph of the introduction the authors mention barriers to DAA still being present in Indonesia. They however failed to state clearly what specific barriers were referring to. The authors will need to clearly state what types of access barriers exist in Indonesia and other lower- and middle-income countries.

3. In the methods section under “Data Collection” the authors list potential predictors of treatment failure without stating how these were determined. If this was based on prior studies this must be stated and appropriately referenced.

4. It is interesting that the study population has 76.6% previous injection drug users with no mention made of current drug users. Did the researchers review the charts for subsequent drug screens in this population to make that determination? Most studies in substance abusing populations suggest that addition is a chronic disease and relapse rates are usually high. If the determination of current drug use was not done in this population the more appropriate description will be individuals with history of IV drug use.

5. The authors failed to state if there were any clinical or other criteria for determining which HCV positive clients were evaluated for HCV treatment. Per the information from the introduction pregnant women and those with Hepatocellular Carcinoma (HCC) were excluded but since pregnancy is not a persistent problem the drop from 64.2% to 42.3% is too great to be due to pregnancy and HCC alone. If other criteria were used to determine who to treat other than HCC and pregnancy such as ongoing drug use, alcohol use etc, this must be included in the manuscript.

6. The authors used the period for designating decimals in most parts of the manuscript but chose to use commas in the data on tolerability of the treatment agents and side effects reported. I suggest they maintain the standard of using a period to designate a decimal here and in all parts of the manuscript.

7. The statement in the results section stating that the observed SVR rate of 27.8% being better than the national rate of 31.3% in coinfected patients is false. Unless I am reading the sentence wrong. The authors must look closely at their SVR numbers and correct their statement appropriately.

8. The authors in their methods and introduction mention performing a logistic regression model to determine predictors of SVR outcomes with DAA treatment. It is however not clear what methods they used to determine the fitness of their model to the data. The criteria used to determine which variables were maintained or excluded from the optimum chosen model was not mentioned. To ensure appropriate reproducibility of the study all these statistical methods will have to be documented.

9. For a predominantly male HIV/HCV coinfected population it is interesting to note a very small proportion of reported same sex HIV/HCV sexual transmission. Is it possible that there are social and cultural factors impacting accurate self-reported HIV risk behaviors?

10. Though the authors did mention some of the possible reasons for the low SVR rate for the cascade mostly at the level of the treatment initiation to SVR that is the stage with the greatest drop. For the most impact on public health and to provide good data for programs seeking to improve the HCV the authors need to focus on each part of the cascade providing possible reasons why individuals failed to transition across the cascade. If those impacted by the program ending are included in the cascade.

11. The authors state as one of the primary objectives determining predictors of HCV treatment outcome with DAA yet this is not mentioned in their conclusion. The authors will need to state in their conclusion which predictors they found to be significant as well as any key covariates determined to have no predictive value.

Reviewer #2: Improving HCV cascade of care is vital to achieve HCV elimination. Even with high rates of SVR, it is important to find predictors of DAAs failure. Authors evaluated these issues in a selected population.

I have some comments:

Why wasn’t the 36% of the population tested for HCV RNA?

You lost many patients in the track: why was that? Can you comment a little more on the causes of failing evaluating fibrosis and initiating treatment? Was the interruption of the program the only motive for not initiating treatment?

In line 149 you wrote: “…but higher SVR12 rate (27.8% vs. 9.5%).” Can you clarify these percentages?

In Predictors of SOF/DCV failure section you are repeating previously presented results. Please correct this.

DBT2 effect might be overestimated due to the small number of patients. The 95%CI is huge (3.2-88.2).

As you mentioned your study has many major limitations:

- Absence of genotyping is a major flaw, specially if GT3 is the second most prevalent in your population since it has a reduce SVR rate to DCV/SOF.

- The small number of patients avoids taking firm conclusions. When analyzing all variables these numbers reduce more.

- This is a very selected population: young age, mostly IVDU, low proportion of cirrhotics, los proportion of previously IFN treated. You must add a comment in the discussion section.

The idea is interesting, especially the analysis of the cascade of care. This can be improved.

You will have to increase the number of patients to take solid conclusion about predictors of failure.

There are some grammatical mistakes. Please, review the text with a professional writer.

6. PLOS authors have the option to publish the peer review history of their article (what does this mean?). If published, this will include your full peer review and any attached files.

Reviewer #1: No

Reviewer #2: No

---

## [Author Response · Author response to Decision Letter 0]

1 Jul 2021

July, 1st, 2021

Dear Editor of Plos One,

 Thank you for the opportunity to revise our manuscript entitled “Hepatitis C Continuum of Care: An Experience of Integrative Hepatitis C Treatment within Human Immunodeficiency Virus Clinic in Indonesia”. The constructive comments from reviewers really helped us to improve content of this manuscript. We sincerely hope that we have addressed the arguments presented and revised the manuscript satisfactorily. 

Reviewer 1 gave a profound evaluation to our manuscript and has brought about important views as well as incisive counterpoints, which we highly appreciate. We also appreciate the detailed correction on the Result Tables and have revised section accordingly. Reviewer 2 provided a brief and comprehensive summary to our research, and we thank the reviewer for the positive response. We have also thoroughly added the study drawbacks in limitation.

One of the authors dr Rahmat Hariyanto passed away due to malignant melanoma at May, 8th 2021. We have added † symbol in author information and a sentence in the acknowledgement.

We have no conflicting interest in writing this manuscript and no financial disclosure need to be done since this study was not received any grant.

We hope that you may find the article in this revised state as satisfactory. We look forward to further feedback or editorial guidance as necessary.

Kind regards,

Evy Yunihastuti

 

Reviewer #1: 

1. The paper provides a good account of a Hepatitis C treatment program in Indonesia. The data presented is good. The authors will however need to revise their manuscript with the help of an editor or a native English speaker to improve the grammar and sentence structure to bring the manuscript up to the standards required for publication.

Thank you for the positive evaluation of our manuscript. We have corrected this new version with the help of professional writer.

2. In the second paragraph of the introduction the authors mention barriers to DAA still being present in Indonesia. They however failed to state clearly what specific barriers were referring to. The authors will need to clearly state what types of access barriers exist in Indonesia and other lower- and middle-income countries.

We sincerely appreciate your suggestion. Several barriers to DAA did persist in lower-middle-income countries, including Indonesia. These barriers include limited case findings, high diagnostic and drug pricing, difficulties in defining the liver fibrosis phase, high patient lost to follow up, weak data systems, and insufficient finances. We stated this description in line 43-45.

3. In the methods section under “Data Collection” the authors list potential predictors of treatment failure without stating how these were determined. If this was based on prior studies this must be stated and appropriately referenced.

Thank you for the detailed correction. We defined the potential predictors based on previous DAAs studies and we have added the references accordingly (line 96).

4. It is interesting that the study population has 76.6% previous injection drug users with no mention made of current drug users. Did the researchers review the charts for subsequent drug screens in this population to make that determination? Most studies in substance abusing populations suggest that addition is a chronic disease and relapse rates are usually high. If the determination of current drug use was not done in this population the more appropriate description will be individuals with history of IV drug use.

Thank you for bringing this issue. The determination of current drug use was not part of standard evaluation before starting DAA at our clinic. Because this study was a retrospective study, we did not have the data. You were right, they are individuals with history of IV drug use (former intravenous drug users). We made changed in Table 1 and add this information in the discussion (line 203-205).

5. The authors failed to state if there were any clinical or other criteria for determining which HCV positive clients were evaluated for HCV treatment. Per the information from the introduction pregnant women and those with Hepatocellular Carcinoma (HCC) were excluded but since pregnancy is not a persistent problem the drop from 64.2% to 42.3% is too great to be due to pregnancy and HCC alone. If other criteria were used to determine who to treat other than HCC and pregnancy such as ongoing drug use, alcohol use etc, this must be included in the manuscript.

Thank you for pointing this out. Pregnant women and hepatocellular carcinoma patients were excluded in the analysis of SOF-DCV treatment failure, not in the continuum of care data. In determining who to threat, for those with severe opportunistic infections or those using rifampicin for tuberculosis treatment, treatment is delayed until the condition is resolved or antituberculosis is stopped. Ongoing drug use and alcohol use were not considered as exclusion criteria to start DAA. We moved these exclusion criteria to line 87-88 and delete the exclusion criteria in the abstract (line 9-10) to avoid confusion.

6. The authors used the period for designating decimals in most parts of the manuscript but chose to use commas in the data on tolerability of the treatment agents and side effects reported. I suggest they maintain the standard of using a period to designate a decimal here and in all parts of the manuscript.

Thank you for reminding us about maintaining the standard format in the whole manuscript. We have changed all the commas to periods in line 126 and 189-190. 

7. The statement in the results section stating that the observed SVR rate of 27.8% being better than the national rate of 31.3% in coinfected patients is false. Unless I am reading the sentence wrong. The authors must look closely at their SVR numbers and correct their statement appropriately.

Thank you for your detailed observation. We humbly apologize for the error in statement 31.3% was the national number of HCV patients who conducted SVR12 evaluation of all patients started treatment, while the number of our study was 88.6%. To avoid the confusion, we changed the paragraph below, comparing DAA initiation and SVR rate only, not comparing SVR VL testing.

Both have shown similar treatment initiation rates: 32.8% of 640 HCV antibody positive patients in this study started DAA treatment while 31.8% of 17200 HCV antibody positive patients in the Indonesian repor started DAA. The SVR12 rate in our study was 27.8%, higher than the SVR12 rate documented in national HCV care cascades (9.5%) (line 182-186).

8. The authors in their methods and introduction mention performing a logistic regression model to determine predictors of SVR outcomes with DAA treatment. It is however not clear what methods they used to determine the fitness of their model to the data. The criteria used to determine which variables were maintained or excluded from the optimum chosen model was not mentioned. To ensure appropriate reproducibility of the study all these statistical methods will have to be documented.

Thank you for detailed observation. We use a backwards stepwise selection process to determine the predictors in multivariate logistic regression. We have added the detailed in line 119-120.

9. For a predominantly male HIV/HCV coinfected population it is interesting to note a very small proportion of reported same sex HIV/HCV sexual transmission. Is it possible that there are social and cultural factors impacting accurate self-reported HIV risk behaviors?

Thank you for bringing this problem. Our clinic was one of the first MSM-friendly clinic in Indonesia. We started with MSM sensitivity training back in 2011, then conducted the training to other HIV centers in Indonesia. Our counselors evaluated all patients risk behaviors before starting antiretroviral therapy. Though it is still possible that the patients did not report accurate HIV risk behavior, we believe the number might be minimal.

10. Though the authors did mention some of the possible reasons for the low SVR rate for the cascade mostly at the level of the treatment initiation to SVR that is the stage with the greatest drop. For the most impact on public health and to provide good data for programs seeking to improve the HCV the authors need to focus on each part of the cascade providing possible reasons why individuals failed to transition across the cascade. If those impacted by the program ending are included in the cascade.

Thank you for the insightful suggestion. We rewrote most of paragraphs in HCV treatment cascade section, trying to explain each part of the cascade and providing possible reasons why individuals failed to transition across the cascade (line 136-147, 150-154, 172-178).

11. The authors state as one of the primary objectives determining predictors of HCV treatment outcome with DAA yet this is not mentioned in their conclusion. The authors will need to state in their conclusion which predictors they found to be significant as well as any key covariates determined to have no predictive value.

Thank you for the suggestion. We have added in the conclusion and abstract that diabetes mellitus was significant predictor of not achieving SVR12 (line 26-27, 273-274).

Reviewer #2: Improving HCV cascade of care is vital to achieve HCV elimination. Even with high rates of SVR, it is important to find predictors of DAAs failure. Authors evaluated these issues in a selected population. 

Thank you for kindly emphasizing the important of the article and the positive overall feedback on the manuscript.

1. Why wasn’t the 36% of the population tested for HCV RNA?

The usual cost for the HCV-RNA test in Indonesia is USD 139 (Trickey, 2019) which is considered too high cost for our patients who mostly come from low-middle-income family. When the government launched the free DAA program, HCV-RNA reagent was included in the coverage. However, supply did not matched the number of anti-HCV-positive patients. Therefore, clinicians might have prioritized HCV-RNA tests for patients who were considered ready for DAA treatment. Some of these patients still had severe opportunistic infections or were using rifampicin for tuberculosis treatment which is known to interact with all available DAA drugs. We have added this discussion in line 141-147.

2. You lost many patients in the track: why was that? Can you comment a little more on the causes of failing evaluating fibrosis and initiating treatment? Was the interruption of the program the only motive for not initiating treatment?

One possible reason for not doing the pre-treatment evaluation is the cost of the diagnostic procedure before starting DAA. Transient elastography (TE) still carries a high-cost, thus often causing a delay in treatment initiation. We have added this information in line 150-152. For those who had pre-treatment evaluation, the interruption of the program was solely the reason not to start treatment.

3. In line 149 you wrote: “…but higher SVR12 rate (27.8% vs. 9.5%).” Can you clarify these percentages?

In our study, 27.8% (178 of 640 anti-HCV positive HIV co-infected patients) achieved SVR12 while in national data 9.5% (1635 of 17200 all HCV-infected patients) achieved SVR12 (Boeke, 2020).

4. In Predictors of SOF/DCV failure section you are repeating previously presented results. Please correct this.

Thank you for the detailed correction. We humbly apologize for this error. We have deleted the repeating results in predictors of SOF-DCV treatment failure section.

5. DBT2 effect might be overestimated due to the small number of patients. The 95%CI is huge (3.2-88.2).

The small number of data did limit our ability to analyze the true association to DBT2 with SOF-DCV treatment failure as the 95% CI was huge. Patel, et al and Romero-Gomez, et al also shown an association of diabetes and insulin resistance to SVR12 achievement. We have added this limitation in line 260-263.

6. As you mentioned your study has many major limitations:

- Absence of genotyping is a major flaw, specially if GT3 is the second most prevalent in your population since it has a reduce SVR rate to DCV/SOF.

- The small number of patients avoids taking firm conclusions. When analyzing all variables these numbers reduce more.

- This is a very selected population: young age, mostly IVDU, low proportion of cirrhotics, los proportion of previously IFN treated. You must add a comment in the discussion section.

Thank you very much for the insightful critique. We have added these limitation details in the discussion (line 259-267).

7. The idea is interesting, especially the analysis of the cascade of care. This can be improved.

Thank you for the crucial suggestion. We rewrote most of paragraphs in HCV treatment cascade section, trying to explain each part of the cascade and providing possible reasons why individuals failed to transition across the cascade (line 136-147, 150-154, 172-178).

8. You will have to increase the number of patients to take solid conclusion about predictors of failure.

Thank you for bringing this important point. It is absolutely the right thing to do when we have more patients. Other studies reporting higher number of patients, but with various DAA regimen, while this study only for sofosbuvir and daclatasvir combination.

9. There are some grammatical mistakes. Please, review the text with a professional writer.

Thank you for the suggestion. We have made some corrections according to professional writer’s suggestion.

---

## [Decision Letter · Decision Letter 1]

13 Jul 2021

PONE-D-21-12487R1

Hepatitis C Continuum of Care: An Experience of Integrative Hepatitis C Treatment within Human Immunodeficiency Virus Clinic in Indonesia

PLOS ONE

Dear Dr. Yunihastuti,

Thank you for submitting your manuscript to PLOS ONE. After careful consideration, we feel that it has merit but does not fully meet PLOS ONE’s publication criteria as it currently stands. Therefore, we invite you to submit a revised version of the manuscript that addresses the points raised during the review process.

Your modified manuscript was reviewed by two original reviewers. Although one reviewer was satisfied with modifications, the other still identified several important problems and made valuable suggestions to improve your paper. Please carefully review the attached comments and provide point-by-point responses.

We look forward to receiving your revised manuscript.

Kind regards,

Yury E Khudyakov, PhD

Academic Editor

PLOS ONE

Reviewers' comments:

Reviewer's Responses to Questions

**Comments to the Author**

1. If the authors have adequately addressed your comments raised in a previous round of review and you feel that this manuscript is now acceptable for publication, you may indicate that here to bypass the “Comments to the Author” section, enter your conflict of interest statement in the “Confidential to Editor” section, and submit your "Accept" recommendation.

Reviewer #1: (No Response)

Reviewer #2: All comments have been addressed

2. Is the manuscript technically sound, and do the data support the conclusions?

Reviewer #1: No

Reviewer #2: Partly

3. Has the statistical analysis been performed appropriately and rigorously? 

Reviewer #1: No

Reviewer #2: Yes

4. Have the authors made all data underlying the findings in their manuscript fully available?

Reviewer #1: Yes

Reviewer #2: Yes

5. Is the manuscript presented in an intelligible fashion and written in standard English?

Reviewer #1: No

Reviewer #2: Yes

6. Review Comments to the Author

Reviewer #1: 1. The Title could be modified to read “Hepatitis C Continuum of Care: Experience of Integrative Hepatitis C Treatment within a Human Immunodeficiency Virus Clinic in Indonesia”, this reads better with same number of words.

2. The sentence on page 15 line 53 “This study was carried out in a specific clinical care of HCV treatment within an integrated HIV clinic in a university hospital in Jakarta” is not clear enough. It is difficult to tell what the authors are attempting to convey with that sentence. This sentence will need to be revise to improve clarity.

3. I suggest the authors rather than use predictors of treatment failure consider the term correlates of treatment failure. The term predictors may suggest some causality, this study is however only able to determine associations and not causality. Correlate is thus a more accurate term for the identified variables in such an analysis.

4. On page 8 and 9 line 127 – 129, I suggest the authors change that sentence suggesting interaction with all available, HCV drugs. This statement is based on mechanism of action some of these interactions have not been formally studied. A more accurate phrase will be “………. most available DAA drugs”.

5. In the manuscript the authors quote a SVR12 rate of 27.8% for their analysis and a 9.5% rate for Indonesia. This numbers appear to be the successful treatment rate from the cascade. Using the term SVR12 which must only be used to describe successful treatment in those who received HCV antivirals is incorrect. The effective SVR12 rate is 84.7%. This is repeated in many parts of the paper the authors need to make sure they correct that error.

6. The reported SVR12 of 95.7% is higher than what I calculated of 84.7% based on all patients who initiated treatment. That is the method used in most analysis, I would suggest the authors clearly define SVR in their analysis since it appears to be different from standard practice. Also, since this calculation is different from other studies comparing with phase III studies and other reports is really not a fair and accurate statement. The authors could do multiple analysis for SVR, but it needs to be clear what the denominator is for each calculation.

7. The manuscript authors will require to work closely with an English language editor to ensure the message provided in this manuscript is accurate to avoid errors in messaging due to difficulties with English.

Reviewer #2: Authors accomplished all suggestions made by the reviewers.

There are still some limitations, but these cannot be corrected (as suggested in the Discussion).

Thanks for your effort and your time.

7. PLOS authors have the option to publish the peer review history of their article (what does this mean?). If published, this will include your full peer review and any attached files.

Reviewer #1: No

Reviewer #2: No

---

## [Author Response · Author response to Decision Letter 1]

22 Jul 2021

July 22nd, 2021

Dear Plos One Editor,

 Thank you for the opportunity to revise our manuscript entitled “Hepatitis C Continuum of Care: Experience of Integrative Hepatitis C Treatment within a Human Immunodeficiency Virus Clinic in Indonesia”. The constructive comments from reviewers really helped us to improve content of this manuscript. 

We have no conflicting interest in writing this manuscript and no financial disclosure need to be done since this study was not received any grant.

We hope that you may find the article in this revised state as satisfactory. We look forward to further feedback or editorial guidance as necessary.

Kind regards,

Evy Yunihastuti

 

Reviewer #1: 

1. The Title could be modified to read “Hepatitis C Continuum of Care: Experience of Integrative Hepatitis C Treatment within a Human Immunodeficiency Virus Clinic in Indonesia”, this reads better with same number of words.

We really appreciate your valuable suggestion. We have changed the title according to your suggestion.

2. The sentence on page 15 line 53 “This study was carried out in a specific clinical care of HCV treatment within an integrated HIV clinic in a university hospital in Jakarta” is not clear enough. It is difficult to tell what the authors are attempting to convey with that sentence. This sentence will need to be revise to improve clarity.

Thank you for pointing this out. We changed this sentence to improve clarity. This study was carried out in HIV integrative clinic of a university hospital in Jakarta. (page 5 line 54)

3. I suggest the authors rather than use predictors of treatment failure consider the term correlates of treatment failure. The term predictors may suggest some causality, this study is however only able to determine associations and not causality. Correlate is thus a more accurate term for the identified variables in such an analysis.

Thank you very much for the insightful critique. We understand that this study was unable to determine causality. Therefore, we have changed the term predictor to factor correlated with treatment failure in the whole manuscript.

4. On page 8 and 9 line 127 – 129, I suggest the authors change that sentence suggesting interaction with all available, HCV drugs. This statement is based on mechanism of action some of these interactions have not been formally studied. A more accurate phrase will be “………. most available DAA drugs”.

Thank you for your detailed observation. We have changed the sentence according to your suggestion. Rifampicin is known to interact with most available DAA drugs. (page 8 line 127-129)

5. In the manuscript the authors quote a SVR12 rate of 27.8% for their analysis and a 9.5% rate for Indonesia. This numbers appear to be the successful treatment rate from the cascade. Using the term SVR12 which must only be used to describe successful treatment in those who received HCV antivirals is incorrect. The effective SVR12 rate is 84.7%. This is repeated in many parts of the paper the authors need to make sure they correct that error.

Thank you for your bringing this point. Studies evaluating HCV continuum of care calculated SVR12 rate in various ways: Boerekamps, et al calculated SVR12 based on patients retained in care; Rizk, et al calculated based on all HCV patients; Saris, et al and Boeke, et al used number of patients with available treatment outcome as denominator; Adekunle, et al used number of patients initiate HCV treatment as denominator of SVR12 rate. We used SVR12 definition the same way as Rizk, et al to define SVR12 rate for the cascade, which was 27.8% (178 of 640 all HIV/HCV co-infected patients). We manually calculated the SVR12 rate for Indonesia in the same way (9.5%: 1635 of 17200 all HCV-infected patients).

When evaluating predictor or SOF-DCV treatment failure, the denominator was all patients using SOF-DCV regimen who had treatment outcome data (176 of 184 patients; 95.7%). Two other patients with available treatment outcome were using SOF-SMV and EBR-GZR regimen, not SOF-DCV. 

We have changed the method section (line 79-96) and added the denominator for each calculation of SVR12 as your suggestion (line. 164-166 and 198)

6. The reported SVR12 of 95.7% is higher than what I calculated of 84.7% based on all patients who initiated treatment. That is the method used in most analysis, I would suggest the authors clearly define SVR in their analysis since it appears to be different from standard practice. Also, since this calculation is different from other studies comparing with phase III studies and other reports is really not a fair and accurate statement. The authors could do multiple analysis for SVR, but it needs to be clear what the denominator is for each calculation.

Thank you for the insightful suggestion. We have added the denominator for each calculation of SVR12 according to your suggestion (line 164-166 and 198). We did two different analysis for SVR12. First, SVR12 rate as part of HCV continuum of care cascade, which used HCV antibody positivity as denominator. Second, when evaluating predictor or SOF-DCV treatment failure, the denominator was all patients using SOF-DCV regimen who had treatment outcome data (176 of 184 patients; 95.7%). This was considered the effective SVR12 of SOF-DCV that can be compared with the results of phase III studies and other reports. 

7. The manuscript authors will require to work closely with an English language editor to ensure the message provided in this manuscript is accurate to avoid errors in messaging due to difficulties with English.

Thank you for your suggestion. We have worked closely with English editor and native speaker for this correction.

Reviewer #2:

Authors accomplished all suggestions made by the reviewers. There are still some limitations, but these cannot be corrected (as suggested in the Discussion). Thanks for your effort and your time.

We appreciate the reviewer for the positive response and all suggestions for our manuscript.

---

## [Editor Report · Decision Letter 2]

2 Aug 2021

Hepatitis C Continuum of Care: Experience of Integrative Hepatitis C Treatment within a Human Immunodeficiency Virus Clinic in Indonesia

PONE-D-21-12487R2

Dear Dr. Yunihastuti,

We’re pleased to inform you that your manuscript has been judged scientifically suitable for publication and will be formally accepted for publication once it meets all outstanding technical requirements.

Kind regards,

Yury E Khudyakov, PhD

Academic Editor

PLOS ONE
---

## [Editor Report · Acceptance letter]

4 Aug 2021

PONE-D-21-12487R2 

Hepatitis C Continuum of Care: Experience of Integrative Hepatitis C Treatment within a Human Immunodeficiency Virus Clinic in Indonesia 

Dear Dr. Yunihastuti:

I'm pleased to inform you that your manuscript has been deemed suitable for publication in PLOS ONE. Congratulations! Your manuscript is now with our production department. 

Kind regards, 

on behalf of

Dr. Yury E Khudyakov 

Academic Editor

PLOS ONE